# Replication Study: The common feature of leukemia-associated IDH1 and IDH2 mutations is a neomorphic enzyme activity converting alpha-ketoglutarate to 2-hydroxyglutarate

Megan Reed Showalter[1†], Jason Hatakeyama[2,3†], Tomas Cajka[1†], Kacey VanderVorst[2,3], Kermit L Carraway III[2,3], Oliver Fiehn[1], Reproducibility Project: Cancer Biology*

[1]West Coast Metabolomics Center, University of California, Davis, United States; [2]Department of Biochemistry and Molecular Medicine, University of California, California, United States; [3]University of California Davis Comprehensive Cancer Center, University of California, California, United States

*For correspondence: tim@cos.io; nicole@scienceexchange.com

[†]These authors contributed equally to this work

Group author details:
Reproducibility Project: Cancer Biology See page 13

**Abstract** In 2016, as part of the Reproducibility Project: Cancer Biology, we published a Registered Report (*Fiehn et al., 2016*), that described how we intended to replicate selected experiments from the paper "The common feature of leukemia-associated IDH1 and IDH2 mutations is a neomorphic enzyme activity converting alpha-ketoglutarate to 2-hydroxyglutarate" (Ward et al., 2010). Here, we report the results of those experiments. We found that cells expressing R172K mutant IDH2 did not display isocitrate-dependent NADPH production above vector control levels, in contrast to the increased production observed with wild-type IDH2. Conversely, expression of R172K mutant IDH2 resulted in increased alpha-ketoglutarate-dependent consumption of NADPH compared to wild-type IDH2 or vector control. These results are similar to those reported in the original study (Figure 2; Ward et al., 2010). Further, expression of R172K mutant IDH2 resulted in increased 2HG levels within cells compared to the background levels observed in wild-type IDH2 and vector control, similar to the original study (Figure 3D; Ward et al., 2010). In primary human AML samples, the 2HG levels observed in samples with mutant *IDH1* or *IDH2* status were higher than those observed in samples without an *IDH* mutation, similar to what was observed in the original study (Figure 5C; Ward et al., 2010). Finally, we report meta-analyses for each result.

## Introduction

The Reproducibility Project: Cancer Biology (RP:CB) is a collaboration between the Center for Open Science and Science Exchange that seeks to address concerns about reproducibility in scientific research by conducting replications of selected experiments from a number of high-profile papers in the field of cancer biology (*Errington et al., 2014*). For each of these papers a Registered Report detailing the proposed experimental designs and protocols for the replications was peer reviewed and published prior to data collection. The present paper is a Replication Study that reports the results of the replication experiments detailed in the Registered Report (*Fiehn et al., 2016*) for a paper by Ward et al., and uses a number of approaches to compare the outcomes of the original experiments and the replications.

In 2010, Ward et al. demonstrated that a shared feature of isocitrate dehydrogenase (IDH) mutations identified in gliomas and acute myeloid leukemia (AML) is the acquisition of a neomorphic enzyme activity. This study built upon a previous study demonstrating cancer-associated *IDH1* mutations resulted in increased production of the oncometabolite 2-hydroxyglutarate (2HG) (*Dang et al., 2009*). Similar to the IDH1 R132 mutation, the cancer-associated IDH2 R172 mutation was demonstrated to be deficient in isocitrate-dependent NADPH production (*Yan et al., 2009*; *Ward et al., 2010*), while acquiring a gain of function for alpha-ketoglutarate-dependent NADPH consumption that resulted in the production of 2HG (*Ward et al., 2010*). 2HG was also reported to be detectable in AML patient samples and predictive of *IDH* mutational status (*Ward et al., 2010*).

The Registered Report for the paper by Ward et al. described the experiments to be replicated (Figures 2, 3D and 5), and summarized the current evidence for these findings (*Fiehn et al., 2016*). Since that publication, additional studies have reported R172 mutant IDH2 exhibiting decreased NADPH production, as well as increased intracellular accumulation of 2HG compared to wild-type IDH2 (*Carbonneau et al., 2016*; *Robinson et al., 2016*). Similarly, there have been numerous studies that have observed this with IDH1 mutations (*Chaturvedi et al., 2017*; *Ward et al., 2013*). There have also been numerous studies reporting the utility of using 2HG as a predictive biomarker for the presence of *IDH* mutations. This includes the detection of 2HG in the peripheral blood of patients with AML (*DiNardo et al., 2013*; *McGehee et al., 2016*), serum samples from patients with a myeloid sarcoma diagnosis (*Willekens et al., 2015*), serum samples from patients with angioimmunoblastic T-cell lymphoma (*Lemonnier et al., 2016*), as well as plasma, urine, and the cerebrospinal fluid in glioma patients (*Fathi et al., 2016*; *Kalinina et al., 2016*; *Lombardi et al., 2015*). Additional strategies have also been reported to detect 2HG noninvasively in glioma that are based on advanced imaging and spectroscopy approaches (*An et al., 2016*; *de la Fuente et al., 2016*; *Ganji et al., 2016*; *Verma et al., 2016*). Clinical trials are also underway to target mutant IDH1 and IDH2 in AML, gliomas, myelodysplastic syndrome, and other solid tumors (*Mondesir et al., 2016*; *Vander Heiden and DeBerardinis, 2017*).

The outcome measures reported in this Replication Study will be aggregated with those from the other Replication Studies to create a dataset that will be examined to provide evidence about reproducibility of cancer biology research, and to identify factors that influence reproducibility more generally.

## Results and discussion

### Assessing the isocitrate-dependent NADPH production and alpha-ketoglutarate-dependent NADPH consumption of wild-type or R172K mutant IDH2

We sought to independently replicate whether the cancer-associated IDH2 R172 mutation, exhibited a loss-of-function for isocitrate-dependent NADPH production and a gain-of-function for alpha-ketoglutarate-dependent NADPH consumption compared to wild-type IDH2. This experiment is similar to what was reported in Figure 2 of *Ward et al. (2010)* and described in Protocol 1 in the Registered Report (*Fiehn et al., 2016*). HEK293T cells, which endogenously express wild-type IDH2, were transfected to ectopically express wild-type or R172 mutant IDH2 (*Figure 1E*). Extracts from cells expressing wild-type IDH2 displayed an increase in isocitrate-dependent NADPH production (0.49 [n = 7, $SD$ = 0.090] change in $OD_{340}$ after 30 min); while, extracts from cells expressing R172K mutant IDH2 reached a 0.12 [n = 7, $SD$ = 0.069] change in $OD_{340}$ over the same period, similar to the levels observed in extracts from vector control transfected cells (0.094 [n = 7, $SD$ = 0.055] change in $OD_{340}$) (*Figure 1A*). This compares to the original study that reported extracts from wild-type IDH2 expressing cells increased isocitrate-dependent NADPH production (~1.3 change in $OD_{340}$ after 30 min), while R172K mutant IDH2 and vector control resulted in an ~0.1 and ~0.02 change in $OD_{340}$, respectively (*Ward et al., 2010*). The same extracts were also tested for alpha-ketoglutarate-dependent NADPH consumption (*Figure 1C*). Expression of R172K mutant IDH2 resulted in a 0.022 [n = 7, $SD$ = 0.0077] change in $OD_{340}$ after 180 min, while wild-type IDH2 transfected cell extracts had minimal activity (0.0034 [n = 7, $SD$ = 0.0019] change in $OD_{340}$), similar to vector control (0.0019 [n = 7, $SD$ = 0.0040] change in $OD_{340}$). This compares to the original study that reported an ~0.06 change in $OD_{340}$ for R172K mutant IDH2 expression, an ~0.014 change in $OD_{340}$ for wild-type IDH2, and an

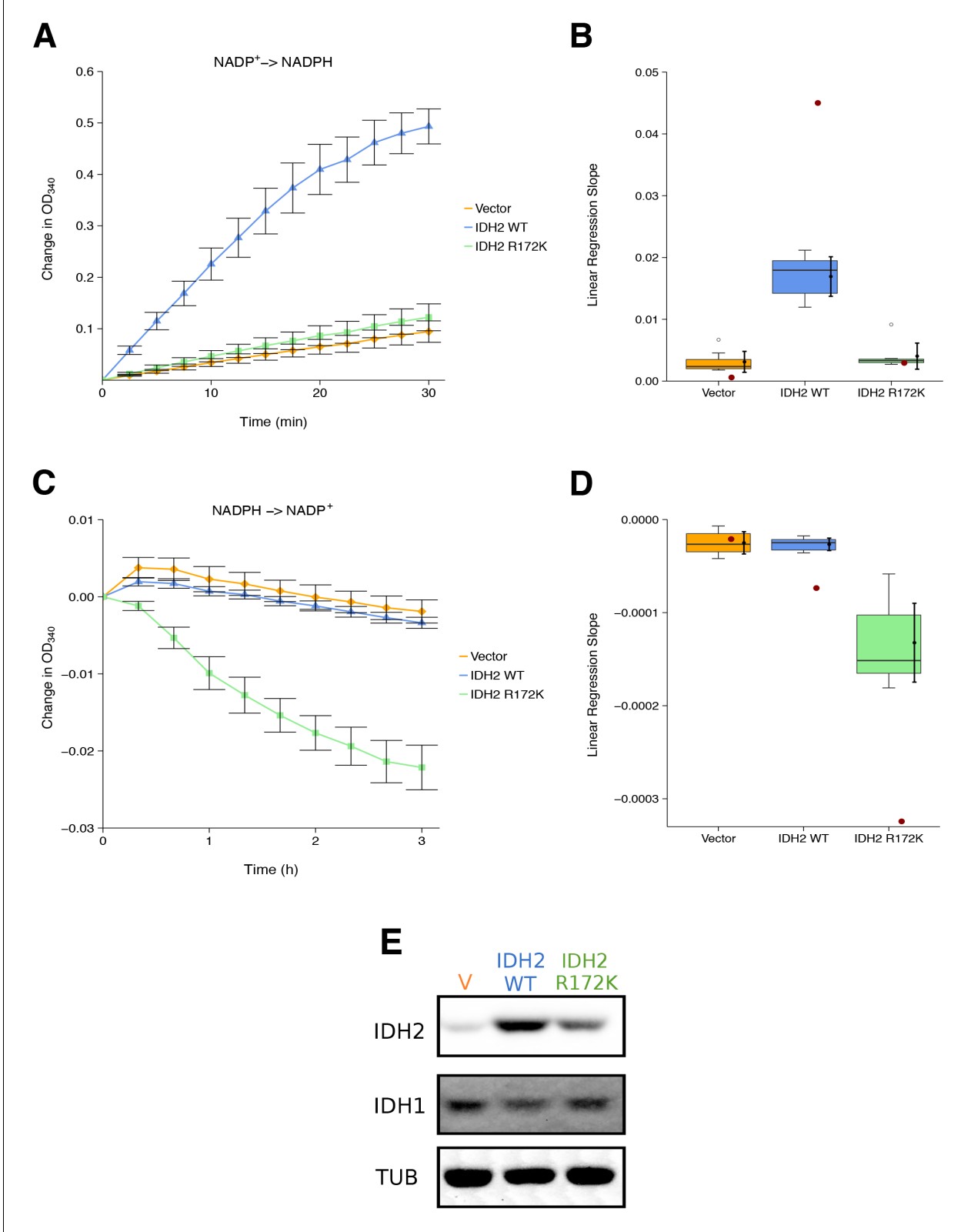

**Figure 1.** Isocitrate-dependent NADPH production and alpha-ketoglutarate-dependent NADPH consumption of wild-type or R172 mutant IDH2. IDH oxidative and reductive activity assays were performed on lysates of HEK293T cells transfected with wild-type or R172K mutant IDH2, or empty vector. (A) Lysates were assessed for generation of NADPH from NADP+ in the presence of 0.4 mM isocitrate over the indicated time course. Mean change in $OD_{340}$ from the beginning of each assay (time = 0) is reported for each biological repeat performed [n = 7] and error bars represent s.e.m. (B) Linear

*Figure 1 continued on next page*

*Figure 1 continued*

regression slopes were determined for each biological repeat of the isocitrate-dependent NADPH production assay. Box and whisker plot with median represented as the line through the box and whiskers representing values within 1.5 IQR of the first and third quartile. Means as black dot and bold error bars represent 95% CI. Linear regression slope determined from the data estimated from the representative experiment reported in Figure 2A of *Ward et al. (2010)* is displayed as a single point (red circle) for comparison. Statistical analysis was performed on data generated during this replication attempt. Wild-type IDH2 compared to vector control: Wilcoxon-Mann-Whitney test; $Z$ = 3.13, uncorrected $p$=0.00058, Bonferroni corrected $p$=0.0023. R172K mutant IDH2 compared to vector control: Wilcoxon-Mann-Whitney test; $Z$ = 1.60, uncorrected $p$=0.13, Bonferroni corrected $p$=0.51. (C) Lysates were assessed for consumption of NADPH in the presence of 1 mM alpha-ketoglutarate over the indicated time course. Mean change in $OD_{340}$ from the beginning of each assay (time = 0) is reported for each biological repeat performed [n = 7] and error bars represent s.e.m. (D) Linear regression slopes were determined for each biological repeat of the alpha-ketoglutarate-dependent NADPH consumption assay. Box and whisker plot with median represented as the line through the box and whiskers representing values within 1.5 IQR of the first and third quartile. Means as black dot and bold error bars represent 95% CI. Linear regression slope determined from the data estimated from the representative experiment reported in Figure 2B of *Ward et al. (2010)* is displayed as a single point (red circle) for comparison. Statistical analysis was performed on data generated during this replication attempt. Wild-type IDH2 compared to vector control: Welch's $t$-test; $t$(9.42) = 0.29, uncorrected $p$=0.78, Bonferroni corrected $p$>0.99. R172K mutant IDH2 compared to vector control: Welch's $t$-test; $t$(6.95) = 5.97, uncorrected $p$=0.00058, Bonferroni corrected $p$=0.0023. (E) Representative Western blots probed with an anti-IDH1 antibody, an anti-IDH2 antibody, and an anti-alpha-Tubulin antibody. Additional details for this experiment can be found at https://osf.io/6ve4d/.

~0.003 change in $OD_{340}$ for vector control after 171 min (*Ward et al., 2010*). The variation in the change in $OD_{340}$ observed between the original and replication studies might be due to methodological differences such as the adjustment in the amount of extract and substrate added to each reaction to generate a sufficient signal to noise, as well as other critical experimental and plate reader settings (*Chavez et al., 2017*; *Giehm and Otzen, 2010*).

There are multiple approaches that could be taken to explore these data; however, to provide a direct comparison to the original data, we are reporting the analysis specified *a priori* in the Registered Report (*Fiehn et al., 2016*). As outlined in the Registered Report, we planned to determine the linear regression slope for each biological repeat and compare vector control to wild-type IDH2 or R172K mutant IDH2 for both assays. To account for these multiple comparisons, the Bonferroni correction was used, making the *a priori* Bonferroni adjusted significance threshold 0.0125. For the isocitrate-dependent NADPH production assay, we performed a test on the slopes from vector control compared to wild-type IDH2, which was statistically significant (Wilcoxon-Mann-Whitney test; $Z$ = 3.13, uncorrected $p$=0.00058, corrected $p$=0.0023). Thus, the null hypothesis that wild-type IDH2 and vector control have similar rates of isocitrate-dependent NADPH production can be rejected, while the test on the slopes from vector control compared to R172K mutant IDH2 was not statistically significant (Wilcoxon-Mann-Whitney test; $Z$ = 1.60, uncorrected $p$=0.13, corrected $p$=0.51). Conversely, for the alpha-ketoglutarate-dependent NADPH consumption assay, the test on the slopes from vector control compared to wild-type IDH2 was not statistically significant (Welch's $t$-test; $t$(9.42) = 0.29, uncorrected $p$=0.78, corrected $p$>0.99), while the test on the slopes from vector control compared to R172K mutant IDH2 was statistically significant (Welch's $t$-test; $t$(6.95) = 5.97, uncorrected $p$=0.00058, corrected $p$=0.0023). To summarize, for this experiment we found results that were in the same direction as the original study and statistically significant where predicted.

## Production of 2HG from wild-type or R172K mutant IDH2 transfected cells

HEK293T cells expressing ectopic wild-type IDH2, R172K mutant IDH2, or vector control were analyzed for accumulation of the metabolite 2HG to test if R172K mutant IDH2 acquired enhanced conversion of alpha-ketoglutarate to 2HG. This experiment is similar to what was reported in Figure 3D of *Ward et al. (2010)*, and described in Protocol 2 in the Registered Report (*Fiehn et al., 2016*), with cells analyzed 24 and 48 hr after transfection. Following expression of the ectopic proteins, the accumulation of 2HG, as well as other organic acids including aspartate and glutamate, were assessed by gas-chromatography mass spectrometry (GC-MS) after MTBSTFA derivatization. Metabolites were identified based on retention times and electron ionization (EI) spectra that matched with derivatized commercial standards. The intracellular levels of 2HG in HEK293T cells expressing R172K mutant IDH2 were determined using the total ion chromatograms (TIC) and found to be over

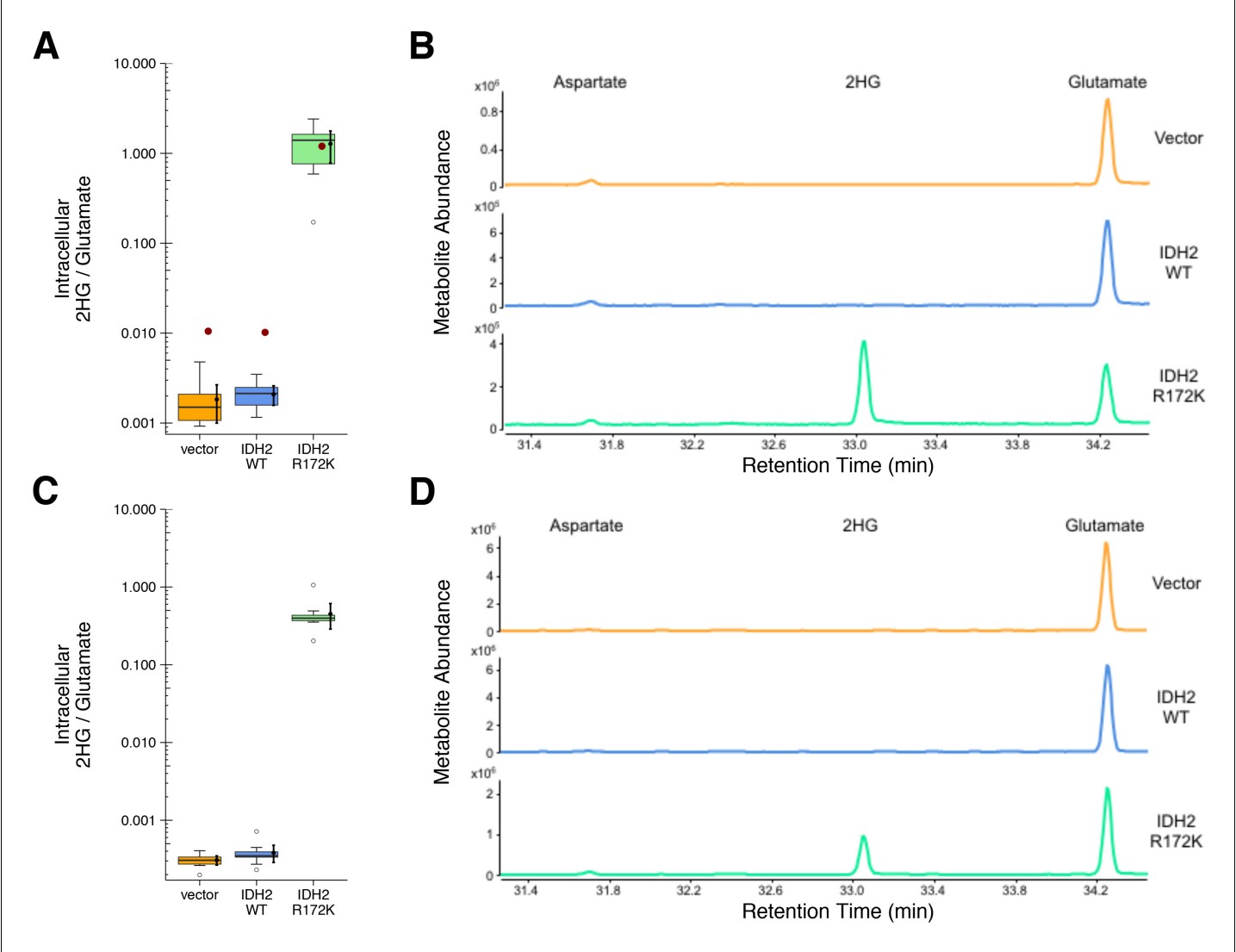

**Figure 2.** 2HG levels within cells expressing wild-type or R172K mutant IDH2. HEK293T cells transfected with wild-type or R172K mutant IDH2, or empty vector, were analyzed for intracellular metabolites. (**A**) Cells harvested 48 hr after transfection had organic acids extracted, purified, and derivatized with MTBSTFA before analysis by GC-MS. Quantitation of 2HG signal intensity relative to glutamate was determined using the TIC for each biological repeat [n = 10]. Box and whisker plot with median represented as the line through the box and whiskers representing values within 1.5 IQR of the first and third quartile. Means as black dot and bold error bars represent 95% CI. Data estimated from the representative experiment reported in Figure 3D of *Ward et al. (2010)* is displayed as a single point (red circle) for comparison. Statistical analysis was performed on $\log_{10}$ transformed data generated during this replication attempt. R172K mutant IDH2 values were compared to the largest value observed in cells expressing vector control or wild-type IDH2 (0.005). One-sample *t*-test; $t(9)$ = 21.9, uncorrected $p$=4.14×10$^{-9}$, Bonferroni corrected $p$=8.28×10$^{-9}$. (**B**) Representative TIC from samples harvested at 48 hr after transfection for vector control (top panel), wild-type IDH2 (middle panel), and R172K mutant IDH2 (bottom panel). The derivatized organic acids eluting between 31.4 and 34.4 min are shown, including aspartate (31.7 min), 2HG (33.1 min), and glutamate (34.3 min) based on spectra of derivatized commercial standards. (**C**) Cells harvested 24 hr after transfection were processed and analyzed similar to the 48 hr samples with [n = 10] biological repeats. Box and whisker plot with median represented as the line through the box and whiskers representing values within 1.5 IQR of the first and third quartile. Means as black dot and bold error bars represent 95% CI. R172K mutant IDH2 values were compared to the largest value observed in cells expressing vector control or wild-type IDH2 (0.001). One-sample Wilcoxon signed-rank test on $\log_{10}$ transformed data; $V$ = 55, uncorrected $p$=0.0020, Bonferroni corrected $p$=0.0039. (**D**) Representative TIC from samples harvested at 24 hr after transfection for vector control (top panel), wild-type IDH2 (middle panel), and R172K mutant IDH2 (bottom panel). Additional details for this experiment can be found at https://osf.io/9ge2a/.

The following figure supplement is available for figure 2:

**Figure supplement 1.** 2HG levels within cells based on EIC analysis.

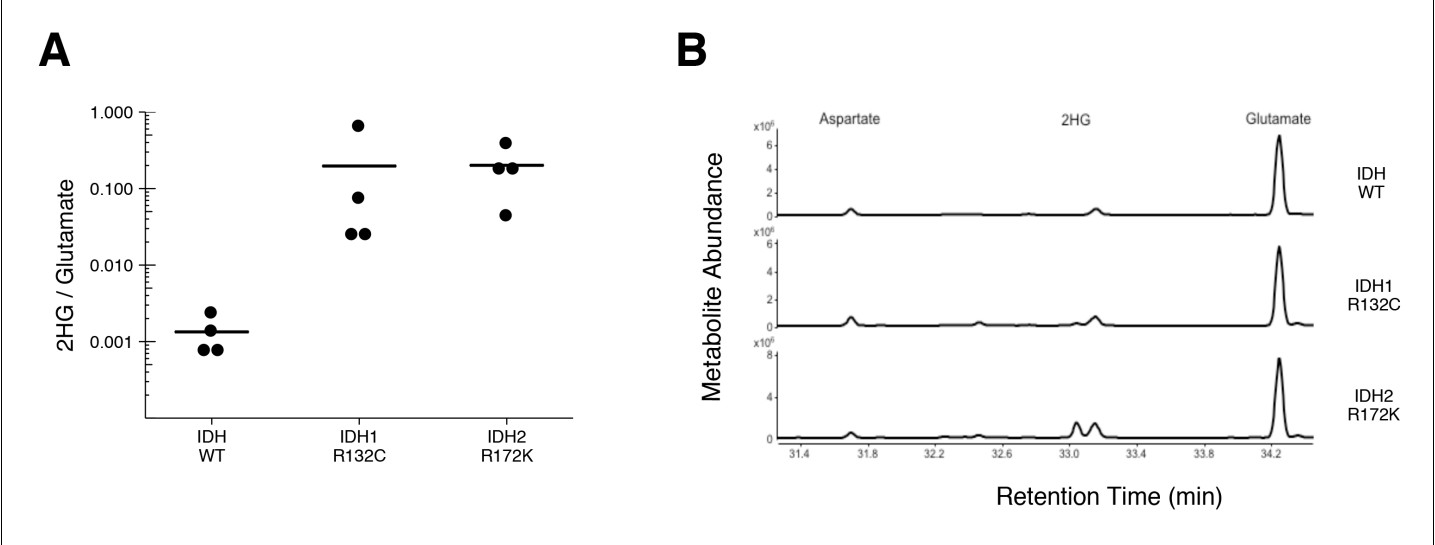

**Figure 3.** 2HG levels in AML patient samples. AML patient peripheral blood or bone marrow samples were analyzed for intracellular metabolites. Cells had organic acids extracted, purified, and derivatized with MTBSTFA before analysis by GC-MS. Patient samples were prescreened for IDH genotypic status prior to metabolite analysis. (**A**) Quantitation of 2HG signal intensity relative to glutamate was determined using the TIC for each sample. Dot plot with means reported as crossbars. One-sample $t$-tests on $\log_{10}$ transformed data comparing 2HG/glutamate level of samples to a constant of 0.0024 (2HG/glutamate threshold between *IDH* mutant and wild-type *IDH* samples). Samples with an *IDH1* mutation compared to the threshold constant: $t(3)$ = 4.47, uncorrected $p$=0.021, Bonferroni corrected $p$=0.063. Samples with an *IDH2* mutation compared to the threshold constant: $t(3)$ = 9.19, uncorrected $p$=0.0027, Bonferroni corrected $p$=0.0082. Samples with an *IDH1* or *IDH2* mutation compared to the threshold constant: $t(7)$ = 8.73, uncorrected $p$=5.20×10$^{-5}$, Bonferroni corrected $p$=1.56×10$^{-4}$. (**B**) Representative TIC from samples without an *IDH* mutation (top panel), samples with an *IDH1* mutation (middle panel), and samples with an *IDH2* mutation (bottom panel). The derivatized organic acids eluting between 31.4 and 34.4 min are shown, including aspartate (31.7 min), 2HG (33.1 min), and glutamate (34.3 min) based on spectra of derivatized commercial standards. Additional details for this experiment can be found at https://osf.io/smdfr/.

The following figure supplement is available for figure 3:

**Figure supplement 1.** 2HG levels in AML patient samples based on EIC analysis.

600 times the levels detected in vector control and wild-type IDH2 expressing cells 48 hr after transfection (*Figure 2A–B*). This compares to the original study, which reported an approximately 100 times increase in intracellular 2HG levels in cells expressing R172K mutant IDH2 compared to vector control and wild-type IDH2 (*Ward et al., 2010*). The difference in magnitude between the original and replication studies is not due to the observed intracellular 2HG/glutamate levels in R172K mutant IDH2 expressing cells, but rather the lower bounds of detection in the vector control and wild-type IDH2 conditions (*Figure 2A–B*). A similar observation was observed 24 hr after transfection as well as analysis using the extracted ion chromatograms (EIC) (*Figure 2C–D*, *Figure 2—figure supplement 1*).

The analysis plan specified in the Registered Report (*Fiehn et al., 2016*) proposed to compare the 2HG/glutamate levels among the different conditions. However, because of the near background levels of vector control and wild-type IDH2, we performed an exploratory analysis using the R172K mutant IDH2 values compared to the largest value observed in cells expressing vector control or wild-type IDH2. This resulted in a statistically significant result for the TIC analysis (one-sample $t$-test; $t(9)$ = 21.9, uncorrected $p$=4.14×10$^{-9}$, Bonferroni corrected $p$=8.28×10$^{-9}$) and the EIC analysis (one-sample $t$-test; $t(9)$ = 22.4, uncorrected $p$=3.38×10$^{-9}$, Bonferroni corrected $p$=6.77×10$^{-9}$) 48 hr after transfection. Similar results were determined on the samples analyzed 24 hr after transfection (one-sample Wilcoxon signed-rank test; $V$ = 55, uncorrected $p$=0.0020, Bonferroni corrected $p$=0.0039). To summarize, for this experiment we found results that were in the same direction as the original study and statistically significant where predicted.

## Assessing 2hg levels in AML patient samples

To test if 2HG is elevated in AML patients with *IDH* mutations, we performed GC-MS analysis on samples of AML cells from patients. This is comparable to the experiment that was reported in Figure 5 of *Ward et al. (2010)*, and described in Protocol 3 in the Registered Report (*Fiehn et al., 2016*), except the mutational status of *IDH1* and *IDH2* were determined prior to conducting the metabolite analysis to have adequate samples for each genetically distinct group. Similar to the original study, all wild-type *IDH* samples displayed a 2HG/glutamate level less than 1%, while samples that had an *IDH1* or *IDH2* mutation had a 2HG/glutamate level greater than 1% (*Figure 3*, *Figure 3—figure supplement 1*). Like the metabolite analysis conducted with HEK293T cells, the lower limits of detection in the patient samples without an *IDH* mutation were lower in this replication attempt (mean 2HG/glutamate ratio of 0.001) than what was reported in the original study (mean 2HG/glutamate ratio of ~0.005). However, the average ratio in the samples with *IDH* mutations reported in this replication attempt (mean 2HG/glutamate ratio of 0.20) were lower than what was reported in the original study (mean 2HG/glutamate ratio of ~0.8). Although these ratios do not allow for a true comparison of the absolute 2HG levels, a comparison of the 2HG peaks between samples with or without *IDH* mutations resulted in a similar trend. Of interest, detection of 2HG in wild-type *IDH* samples, albeit at much lower levels than in samples with an *IDH* mutation, has also been reported in other studies measuring 2HG levels in patient samples (*Borger et al., 2014*; *DiNardo et al., 2013*; *Fathi et al., 2012*; *Gross et al., 2010*; *Janin et al., 2014*; *Lemonnier et al., 2016*; *Natsumeda et al., 2014*). Cells without *IDH* mutations produce small quantities of 2HG, although the mechanism is poorly understood and thought to be largely due to error of normal metabolism (*Andronesi et al., 2013*; *Intlekofer et al., 2015*). While detection of 2HG levels in serum and urine in AML and intrahepatic cholangiocarcinoma has been reported by various studies as a biomarker for *IDH* mutational status, detection in glioma samples does not appear as straightforward with no difference between serum samples, likely due to the blood–brain barrier, and further evaluation needed with the use of urinary samples (*Fathi et al., 2016*; *Lombardi et al., 2016*, *2015*).

As outlined in the Registered Report (*Fiehn et al., 2016*), we planned to conduct three comparisons using the Bonferroni correction to adjust for multiple comparisons, making the *a priori* Bonferroni adjusted significance threshold 0.0167. We performed one-sample *t*-tests comparing the 2HG/glutamate level determined by TIC ($\log_{10}$ transformed) to a constant of 0.0024, which was the 2HG/glutamate threshold to delineate between *IDH* mutant and wild-type *IDH* samples. The sample size was determined *a priori* to detect the effects based on the originally reported data, which reported the 2HG/glutamate threshold as 0.01. The test on samples with an *IDH2* mutation was statistically significant ($t(3) = 9.19$, uncorrected p=0.0027, corrected p=0.0082), as was the test on samples with an *IDH1* or *IDH2* mutation ($t(7) = 8.73$, uncorrected p=$5.20 \times 10^{-5}$, corrected p=$1.56 \times 10^{-4}$). The planned comparison of samples with an *IDH1* mutation, however, was not statistically significant ($t(3) = 4.47$, uncorrected p=0.021, corrected p=0.063). The same analysis was conducted using the 2HG/glutamate levels determined by EIC, where the observed 2HG/glutamate threshold between *IDH* mutant and wild-type *IDH* samples was 0.001. All effects were found to be statistically significant (*IDH1*: $t(3) = 5.40$, uncorrected p=0.012, corrected p=0.037; *IDH2*: $t(3) = 10.93$, uncorrected p=0.0016, corrected p=0.0049; *IDH1/IDH2*: $t(7) = 10.44$, uncorrected p=$1.61 \times 10^{-5}$, corrected p=$4.84 \times 10^{-5}$). To summarize, for this experiment we found results that were in the same direction as the original study and statistically significant, except on samples with *IDH1* mutations when using the same type of chromatograms as the original study. As with all statistical significance testing, these results should take into consideration that whether a result is statistically significant or not is dependent not only on the significance threshold, but also the number of samples (*Sullivan and Feinn, 2012*).

## Meta-analyses of original and replication effects

We performed a meta-analysis using a random-effects model, where possible, to combine each of the effects described above as pre-specified in the confirmatory analysis plan (*Fiehn et al., 2016*). To provide a standardized measure of the effect, a common effect size was calculated for each effect from the original and replication studies. For a one-sample test, Cohen's *d* is the difference between the sample mean and the null value using the sample standard deviation. The estimate of the effect size of one study, as well as the associated uncertainty (i.e. confidence interval), compared to the

effect size of the other study provides another approach to compare the original and replication results (*Errington et al., 2014*; *Valentine et al., 2011*). Importantly, the width of the confidence interval for each study is a reflection of not only the confidence level (e.g. 95%), but also variability of the sample (e.g. *SD*) and sample size.

To compare the original and replication results for the enzymatic assays, the linear regression slope was determined for the values estimated from the representative experiment reported in Figure 2 of *Ward et al. (2010)* and compared to the slopes reported in this replication attempt. For the isocitrate-dependent NADPH production assay, the slopes for wild-type IDH2 and vector control from the original study were outside the 95% confidence interval (CI) of the slopes generated during the replication attempt, while the slope for R172K mutant IDH2 from the original study was within the 95% CI of the replication attempt (*Figure 1B*). For the alpha-ketoglutarate-dependent NADPH consumption assay, the slopes for R172K mutant IDH2 and wild-type IDH2 from the original study fell outside the 95% CI of the slopes generated during the replication attempt, while the slope for vector control from the original study was within the 95% CI of the replication attempt (*Figure 1D*).

Similar to the enzymatic assays, the intracellular 2HG/glutamate levels determined 48 hr after transfection in the original study were estimated from the representative image reported in Figure 3D of *Ward et al. (2010)* and compared to the values reported in this replication attempt. For R172K mutant IDH2, the intracellular 2HG/glutamate levels in the original study were within the 95% CI of the replication attempt, while wild-type IDH2 and vector control fell outside the 95% CI (*Figure 2A*).

There were three comparisons made with the AML patient sample data analyzing 2HG/glutamate levels based on *IDH* mutational status (*Figure 4*). For each of the comparisons, the 2HG/glutamate levels were compared to a constant representing the 2HG/glutamate threshold between IDH mutant and wild-type IDH samples detected in the study (original: 0.01; replication: 0.0024). The comparison of samples with an *IDH1* mutation resulted in $d = 8.40$, 95% CI [2.84, 14.09] for the data estimated *a priori* from Figure 5C in the original study (*Ward et al., 2010*), whereas $d = 2.23$, 95% CI [0.26, 4.17] for this study. A meta-analysis of these two effects resulted in $d = 4.80$, 95% CI [−1.16, 10.76], p=0.114. The original and replication effects were in the same direction, however, the point estimate of the replication effect size was not within the confidence interval of the original result, and vice versa. Further, the large confidence intervals of the meta-analysis along with a statistically significant Cochran's *Q* test (p=0.031), suggests heterogeneity between the original and replication studies. The test of samples with an *IDH2* mutation resulted in $d = 6.75$, 95% CI [3.45, 10.04] for the original study, $d = 4.60$, 95% CI [1.05, 8.22] for this study, with a meta-analysis resulting in $d = 5.72$, 95% CI [3.42, 8.02], p=$1.12 \times 10^{-6}$. Similarly, the test of samples with an *IDH1* or *IDH2* mutation resulted in $d = 4.36$, 95% CI [2.62, 6.09] for the original study, $d = 3.09$, 95% CI [1.35, 4.80] for this study, with a meta-analysis resulting in $d = 3.71$, 95% CI [2.46, 4.96], p=$5.73 \times 10^{-9}$. For each of these two comparisons, the original and replication effects were in the same direction and the point estimate of the replication effect size was within the confidence interval of the original result, and vice versa.

This direct replication provides an opportunity to understand the present evidence of these effects. Any known differences, including reagents and protocol differences, were identified prior to conducting the experimental work and described in the Registered Report (*Fiehn et al., 2016*). However, this is limited to what was obtainable from the original paper, which means there might be particular features of the original experimental protocol that could be critical, but unidentified. So while some aspects, such as cell line, transfection reagent, and metabolite extraction technique were maintained, others were unknown or not easily controlled for. These include variables such as cell line genetic drift (*Hughes et al., 2007*; *Kleensang et al., 2016*), GC-MS parameters such as injection volume (*Allwood et al., 2009*), and other patient characteristics (*Klevorn and Teague, 2016*). Furthermore, recent studies since *Ward et al. (2010)* was published have identified numerous factors, such as hypoxic microenvironments that promote glutamine consumption for glutaminolysis and reductive carboxylation pathways, affecting IDH reactions, specifically alpha-ketoglutarate (*Parker and Metallo, 2015*). Additionally, enantiomers of 2HG have been found to be produced by different enzymes with distinct metabolisms (*Losman and Kaelin, 2013*). Whether these or other factors influence the outcomes of this study is open to hypothesizing and further investigation, which is facilitated by direct replications and transparent reporting.

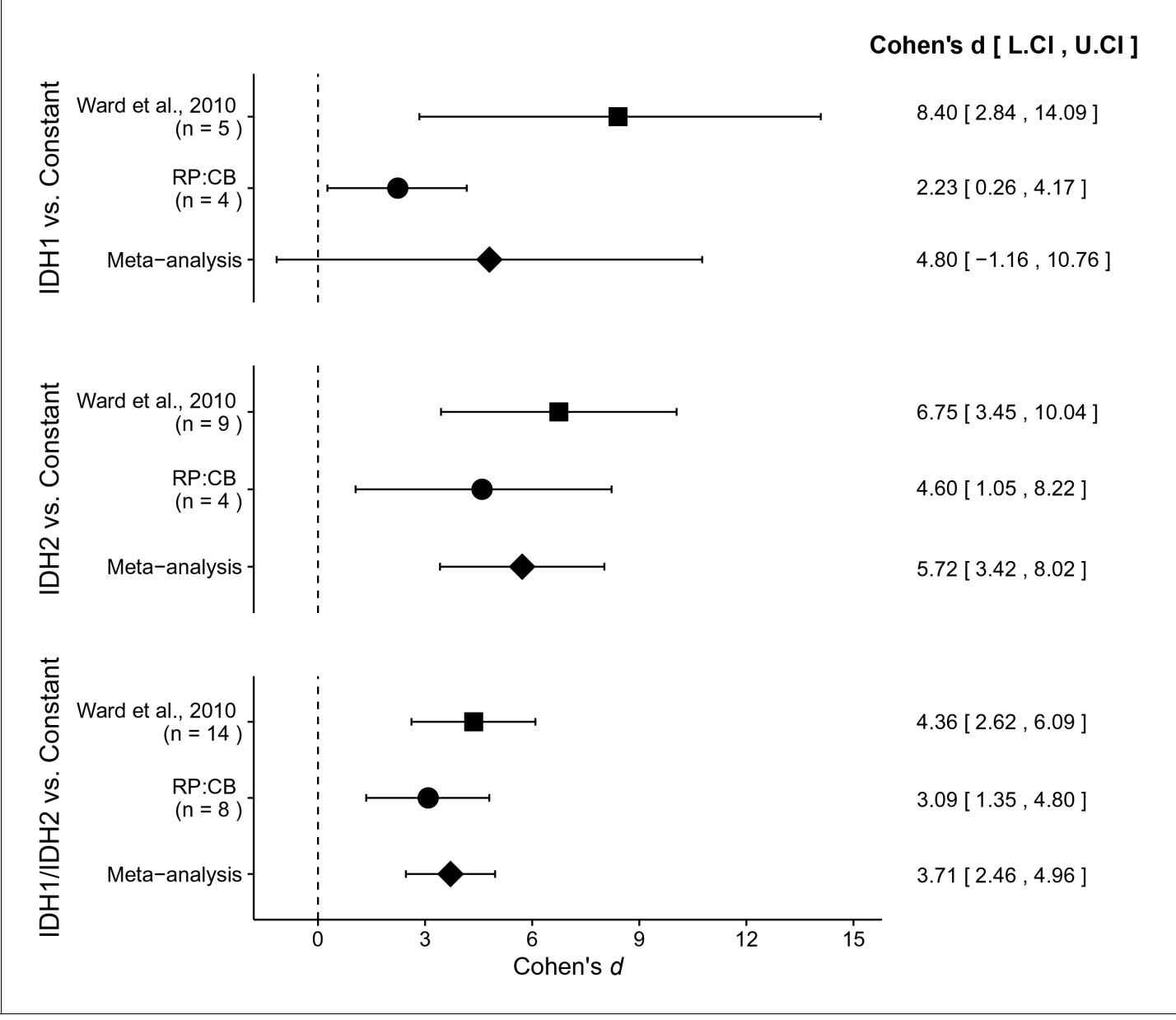

**Figure 4.** Meta-analyses of each effect. Effect size and 95% confidence interval are presented for *Ward et al. (2010)*, this replication study (RP:CB), and a random effects meta-analysis of those two effects. Sample sizes used in *Ward et al. (2010)* and this replication attempt are reported under the study name. Random effects meta-analysis of AML patient samples with an *IDH1* mutation compared to a constant representing the 2HG/glutamate threshold between *IDH* mutant and wild-type *IDH* samples detected in the study (original: 0.01; replication: 0.0024) (meta-analysis $p$=0.114), AML patient samples with an *IDH2* mutation compared to the threshold constant (meta-analysis $p$=1.12×10$^{-6}$), and AML patient samples with an *IDH1* or *IDH2* mutation compared to the threshold constant (meta-analysis $p$=5.73×10$^{-9}$). Additional details for these meta-analyses can be found at https://osf.io/4m3n8/.

## Materials and methods

As described in the Registered Report (*Fiehn et al., 2016*), we attempted a replication of the experiments reported in Figures 2, 3D and 5 of *Ward et al. (2010)*. A detailed description of all protocols can be found in the Registered Report (*Fiehn et al., 2016*). Additional detailed experimental notes, data, and analysis are available on the Open Science Framework (OSF) (RRID:SCR_003238) (https://osf.io/8l4ea/; *Showalter et al., 2017*).

## Cell culture

HEK239T cells (ATCC, cat# CRL-3216, RRID:CVCL_0063) were grown in DMEM supplemented with 10% Fetal Bovine Serum (FBS) (Hyclone, Logan, UT, cat# SH30071.03). Cells were grown at 37°C in a humidified atmosphere at 10% $CO_2$. Quality control data for the cell line are available at https://osf.io/zt996/. This includes results confirming the cell line was free of mycoplasma contamination (MycoAlert PLUS, Lonza, Basel, Switzerland, cat# LT07-701). Additionally, STR DNA profiling of the cell line was performed (DDC Medical, Fairfield, Ohio) and cells were confirmed to be the indicated cell line when queried against STR profile databases.

## Plasmids and transfection

To generate pcDNA3.1-IDH2$^{WT}$ (deposited in Addgene, plasmid# 87926) and pcDNA3.1-IDH2$^{R172K}$ (deposited in Addgene, plasmid# 87927), IDH2$^{WT}$ and IDH2$^{R172K}$ were amplified by PCR from pCMV6-Entry vectors (Origene, Rockville, MD, cat# RC201152 and cat# RC400103) and cloned into the BamH1 and EcoR1 sites of the pcDNA3.1 expression vector (Invitrogen, Carlsbad, CA, cat# V790-20). Identity of all vectors were confirmed by sequencing and agarose gel electrophoresis (https://osf.io/zqyyq/). The day before transfection HEK293T cells were plated and allowed to grow overnight. For the assessment of alpha-ketoglutarate dependent NADPH consumption (*Figure 1*), 6 × 10$^5$ cells were seeded per well in six-well plates. For assessing the production of 2HG (*Figure 2*), 3.5 × 10$^6$ cells were seeded per 10 cm plate. Cells were transfected with the appropriate plasmid with Lipofectamine 2000 (Invitrogen, cat# 11668027) according to manufacturer's instructions. For a 6-well plate, each well was transfected with 1 µg DNA diluted first in 50 µl serum-free, antibiotic-free DMEM and incubated for 3–5 min, and 2.67 µl Lipofectamine 2000 diluted first in 50 µl serum-free, antibiotic-free DMEM and incubated for 3–5 min. Transfections for 10 cm plates included 7.5 µg DNA diluted in 250 µl serum-free, antibiotic-free DMEM, and 20 µl Lipofectamine 2000 diluted in 250 µl serum-free, antibiotic-free DMEM. Following the individual incubations, the diluted DNA and Lipofectamine 2000 was combined and incubated for an additional 10 min before being added to the cells. Medium was replaced 24 hr later.

## Cell lysate based enzyme assays

For IDH oxidative and reductive activity assays, transfected cells were harvested 48 hr after transfection in 0.5 ml/well with mammalian protein extraction reagent (mPER) containing protease and phosphatase inhibitor cocktail (4 µg/ml aprotinin, 0.2 mM AEBSF, 4 µg/ml leupeptin, 4 µg/ml pepstatin A, 0.2 mM NaOV, and 1 mM NaF) on ice. Lysate was collected, homogenized using a T-10 homogenizer with S10N-5 tip (IKA, Staufen im Breisgau, Germany) for 1–2 min on ice, and then centrifuged at 14,000x*g* for 10 min at 4°C. Protein concentration was measured using the DC Protein Assay Kit II (BioRad, Hercules, CA, cat# 5000112) according to manufacturer's instructions. Lysate was aliquoted for IDH oxidative and reductive activity assays and Western blotting. IDH oxidative activity was determined by mixing 15 µg of each protein lysate with 200 µl of assay buffer (100 mM Tris-HCl, pH 7.5, 1.3 mM MnCl2, 0.33 mM EDTA, 0.4 mM ß-NADP$^+$, 0.4 mM D-(+)-threo-isocitrate). Triplicate reactions were measured every 2.5 min for 30 min at 340 nm absorbance using a microplate spectrophotometer (Molecular Devices, Sunnyvale, CA, FilterMax F5 Multi-Mode Microplate Reader) and Multi-Mode Analysis Software (Molecular Devices, RRID:SCR_014789), version 3.4.0.25. IDH reductive activity was determined by mixing 30 µg of each protein lysate with 200 µl of assay buffer (100 mM Tris-HCl, pH 7.5, 1.3 mM MnCl$_2$, 0.02 mM ß-NADPH, 1 mM alpha-ketoglutarate). Triplicate reactions were measured every 20 min for 3 hr at 340 nm. For all experiments, a mPER only sample lacking lysate protein was mixed with assay buffer to determine the background, which was subtracted from the average triplicate reading for each timepoint. For each biological repeat, the value at time 0 was subtracted from all timepoints to calculate change in absorbance.

## Western blot

Sample buffer (50 mM Tris-HCl, pH 6.8, 2% SDS, 10% glycerol, 1% ß-mercaptoethanol, 0.02% bromophenol blue) was added to protein lysates and incubated at 95°C for 5 min. 40 µg of protein along with protein ladder was resolved by SDS-PAGE and transferred to nitrocellulose membrane as described in the Registered Report (*Fiehn et al., 2016*). Membrane was washed with deionized water and Ponceau stain was used to confirm transfer. The membrane was blocked with 5% w/v

nonfat dry milk/0.2% sodium azide in 1X TBS with 0.1% Tween-20 (TBST) for 30 min at room temperature. Incubation with primary antibody was conducted overnight at 4°C. Membranes were probed with goat anti-IDH1 (Santa Cruz Biotechnology, Dallas, TX, cat# sc49996, RRID:AB_2123166); 1:500 dilution in 5% w/v nonfat dry milk/TBST, mouse anti-IDH2 (Abcam, Cambridge, UK, cat# ab55271, RRID:AB_943793); 0.5 μg/ml (1:1,000) dilution in 5% w/v nonfat dry milk/TBST, and mouse anti-alpha-Tubulin (Sigma-Aldrich, St. Louis, MO, cat# T5168, RRID:AB_477579); 1:10,000 dilution in 5% w/v nonfat dry milk/TBST. Each incubation was followed by washes with TBST and the appropriate secondary antibody: HRP-conjugated rabbit anti-goat (Invitrogen, cat# 81–1620, RRID:AB_2534006); 1:5000 dilution in 5% w/v nonfat dry milk/TBST or HRP-conjugated goat anti-mouse (Bio-Rad, cat# 170–5047, RRID:AB_11125753); 1:7500 dilution in 5% w/v nonfat dry milk/TBST. The HRP-conjugated anti-Actin antibody (Cell Signaling, Danvers, MA, cat# 12620) listed in the Registered Report was tried; however, it overlapped with IDH2 and thus could not be utilized. Membranes were washed with TBST and incubated with SuperSignal West Femto Maximum Sensitivity Substrate (Thermo Fisher Scientific, Waltham, MA, cat# 34096) according to the manufacturer's instructions. Western blots were visualized using an Alpha Innotech (San Leandro, CA) imaging station (FluorChem FC2 system) using the AlphaView software (RRID:SCR_014549), version 3.2.2. Between antibody incubations, HRP was inactivated by incubating with 1 mM sodium azide and subsequently imaged to confirm HRP inactivation before proceeding to the next antibody incubation. All images taken are available at https://osf.io/th3za/.

## AML patient samples

Patient samples were obtained from Roswell Park Cancer Institute. The samples were prescreened for *IDH* genotypic status and the pre-specified sample of each group was determined prior to conducting the study. Approval was obtained from the institutional review board (protocol # CIC 9805) and shared samples and data were de-identified for this study. Cells used for this study were prepared by Ficoll separation of mononuclear cells from peripheral blood or bone marrow and were frozen as viable cells in 10% dimethyl sulfoxide (DMSO).

## Metabolite extraction

HEK293T cells were harvested 24 and 48 hr after transfection by removal of medium, while peripheral blood mononuclear cells from frozen AML samples, were centrifuged at 1000x*g* to remove freezing medium. After removal of medium, cells were rapidly quenched with 1.5 ml of methanol chilled to −80°C. Cells were incubated at −80°C for 15 min and in the case of HEK293T cells scraped off the dish. Cells were centrifuged for 5 min at 2000x*g* at 4°C to pellet cellular debris. Pellets were re-extracted by an addition of 500 μl of 80% methanol in water, chilled to −80°C, vortexed, then incubated at 4°C for 15 min and centrifuged for 5 min at 1000x*g* at 4°C.

Supernatants from the multiple extractions were combined and evaporated to dryness using a cold trap concentrator (Labconco, Kansas City, MO). Samples were then resuspended in 200 μl LC-MS grade water. A 2 ml AG-1 × 8 100–200 anion exchange column (Bio-Rad, cat# 731–6211) was washed with five column volumes (10 ml) of 3 N HCl followed by transfer of resuspended extracts to resin column. Metabolites were eluted using 10 ml 3 N HCl. Samples were evaporated to dryness using a cold trap concentrator and resuspended in 100 μl MTBSTFA/ACN (1:1, v/v) and shaken at 60°C for 1 hr using an Orbital Mixing Chilling/Heating Plate (Torrey Pines Scientific Inc., Carlsbad, CA) at a maximum orbital speed of 9. Derivatized extracts were further diluted (1:4) with a MTBSTFA/ACN (1:1, v/v) mixture and transferred to glass vials with micro-inserts and capped immediately prior to GC-MS analysis.

## GC-MS analysis

The system consisted of a GC-MS system (Agilent Technologies, Santa Clara, California), a 7693 Series autosampler, a split/splitless injector, a 7890A GC system, and a quadrupole mass spectrometer 5977A. Injection parameters were as follows: injection volume, 0.2 μl; injector temperature, 250°C; helium carrier gas flow, 1 ml/min; splitless period, 1 min. For GC separation a 30 m x 0.25 mm, 0.25 μm HP-5MS UI (Agilent Technologies) capillary column was used with an oven temperature program: 100°C (3 min), 4 °C/min to 230°C (hold 4 min), 30 °C/min to 300°C (hold 5 min). MS detection parameters were as follows: electron ionization, −70 eV; acquisition rate, 2.71 scans/s; mass

range, *m/z* 50–600; MS ion source temperature, 230°C; MS quadrupole temperature, 150°C; electron multiplier voltage, 1651 V. For the data acquisition, the MSD Productivity ChemStation Software (Agilent Technologies), version E.02.00 was used. Method blanks and quality control samples were run with experimental samples for quality control purposes. Both total ion chromatograms (TIC) and extracted ion chromatograms (EIC) are reported. Data files were processed using MassHunter Qualitative Analysis Software (Agilent Technologies), version B.05.00 and MassHunter Quantitative Analysis Software (Agilent Technologies), version B.05.01. Retention times for compounds were as follows: 31.7 min aspartate, 34.3 min glutamate, and 33.1 min 2-hydroxyglutarate. Metabolites were identified based on retention times and electron ionization (EI) spectra matching with authentic standards (Sigma, cat# 2HG:H8378, Asp:A9256, Glu:G1251). This metabolomics data is available at the NIH Common Fund's Data Repository and Coordinating Center (supported by NIH grant, U01-DK097430) website, (http://www.metabolomicsworkbench.org), where it has been assigned a Metabolomics Workbench Project ID: ST000548. The data is directly accessible at http://www.metabolomicsworkbench.org/data/DRCCMetadata.php?Mode=Study&StudyID=ST000548.

## Statistical analysis

Statistical analysis was performed with R software (RRID:SCR_001905), version 3.3.2 (*R Core Team, 2016*). All data, csv files, and analysis scripts are available on the OSF (https://osf.io/8l4ea/). Confirmatory statistical analysis was pre-registered (https://osf.io/t873r/) before the experimental work began as outlined in the Registered Report (*Fiehn et al., 2016*). Proposed analysis of GC-MS data was conducted on TIC data, with additional exploratory analysis performed using the EIC data. Data were checked to ensure assumptions of statistical tests were met. When described in the results, the Bonferroni correction, to account for multiple testings, was applied to the alpha error or the *p*-value. The Bonferroni corrected value was determined by dividing the uncorrected value (.05) by the number of tests performed. Although the Bonferroni method is conservative, it was accounted for in the power calculations to ensure sample size was sufficient. A meta-analysis of a common original and replication effect size was performed with a random effects model and the *metafor* package (*Viechtbauer, 2010*) (available at: https://osf.io/4m3n8/). The original study data was extracted *a priori* from the published figures by estimating the value reported. The extracted data from Figures 2A-B of *Ward et al. (2010)* were used to calculate the mean slope after fitting the data to a linear regression. The extracted data were published in the Registered Report (*Fiehn et al., 2016*) and was used in the power calculations to determine the sample size for this study.

## Deviations from registered report

The expression vector pcDNA3.1 was used instead of pcDNA3 since it was unavailable for order at the time the study was conducted. The concentrations of protein used was increased for the IDH oxidative (0.3 µg to 15 µg) and reductive (3 µg to 30 µg) assays, as well as the amount of substrate (D-(+)-threo-isocitrate: 0.1 mM to 0.4 mM; alpha-ketoglutarate: 0.5 mM to 1 mM). This was due to an inability to detect a signal above the noise with the originally specified protein concentrations, with the need to increase the substrate concentration due to saturation before the end of the time points. Cell lysates were also homogenized instead of sonicated to generate protein extracts. Following this optimization, the same procedure was carried out for the biological repeats reported. For the Western blot, Actin was not used as the additional control since it overlapped with IDH2; Alpha-Tubulin was used instead. For the GC-MS, the Registered Report listed MTBSTFA in the materials, but incorrectly listed MSTFA in the procedure. MTBSTFA was used in this replication attempt, similar to the original study. For the GC-MS run, it was unstated how ramping to 300°C was to be performed, as such 30 °C/min was used. Derivatized extracts were diluted 1:4 to ensure metabolites were in proper concentration range for the GC-MS instrument used since the cell number, derivatization volume, and injection volume were not listed in the original study. Additional materials and instrumentation not listed in the Registered Report, but needed during experimentation are also listed.

## Acknowledgements

The Reproducibility Project: Cancer Biology would like to thank Courtney Soderberg at the Center for Open Science for assistance with statistical analyses and the following companies for generously

donating reagents to the Reproducibility Project: Cancer Biology; American Type and Tissue Collection (ATCC), Applied Biological Materials, BioLegend, Charles River Laboratories, Corning Incorporated, DDC Medical, EMD Millipore, Harlan Laboratories, LI-COR Biosciences, Mirus Bio, Novus Biologicals, Sigma-Aldrich, and System Biosciences (SBI).

## Additional information

### Group author details

Reproducibility Project: Cancer Biology

Elizabeth Iorns: Science Exchange, Palo Alto, United States; Alexandria Denis: Center for Open Science, Charlottesville, United States; Nicole Perfito: Science Exchange, Palo Alto, California; Timothy M Errington, http://orcid.org/0000-0002-4959-5143: Center for Open Science, Charlottesville, United States

### Competing interests

MRS, TC, OF: West Coast Metabolomics Center, University of California, Davis is a Science Exchange associated lab. RP:CB: EI, NP: Employed by and hold shares in Science Exchange Inc. The other authors declare that no competing interests exist.

### Funding

| Funder | Author |
| --- | --- |
| Laura and John Arnold Foundation | Reproducibility Project: Cancer Biology |

The funder had no role in study design, data collection and interpretation, or the decision to submit the work for publication.

### Author contributions

MRS, JH, TC, KV, KLC, RP:CB, OF, Acquisition of data, Drafting or revising the article

### Author ORCIDs

Jason Hatakeyama, http://orcid.org/0000-0001-8690-5107

### Ethics

Human subjects: Patient samples were obtained from Roswell Park Cancer Institute. The samples were prescreened for IDH genotypic status and the pre-specified sample of each group was determined prior to conducting the study. Approval was obtained from the institutional review board (protocol # CIC 9805) and shared samples and data were de-identified for this study.

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
