## [Decision Letter]

Thank you for submitting your article "Replication Study: The common feature of leukemia-associated IDH1 and IDH2 mutations is a neomorphic enzyme activity converting α-ketoglutarate to 2-hydroxyglutarate" for consideration by *eLife*. Your article has been reviewed by two peer reviewers, and the evaluation has been overseen by a Reviewing Editor and Vivek Malhotra as the Senior Editor. The reviewers have opted to remain anonymous.

The reviewers have discussed the reviews with one another and the Reviewing Editor has drafted this decision to help you prepare a revised submission.

Summary

The inherent property of any reproducibility study is that its design is directly related to the original work and set of experiments and cannot extend beyond. This exact situation has arisen with the current reproducibility study, that has been conducted according to the approved plan and results were carefully taken and were adequately statistically treated. It is not surprising that with exception of certain absolute values which did not match to the original publication of Ward et al., the results were reproduced with a high accuracy. The methodical approach is excellent and of the top level, adequate to the task, including derivatization and temperature gradients. Overall, the experiments appear to be performed to a high standard and clearly reproduce the majority of findings from the Ward paper, which is expected based on other independently published studies.

Essential revisions

1) It should be pointed out that specifically precleaning of samples on a column adds to a straightforward analysis. However, it should be verified, whether such precleaning does not withdraw from the sample any desired species to be tested.

2) To bring an additional value besides the simple fact of reproducibility, the authors should discuss why certain other studies have found non zero levels of 2-hydroxyglutarate in cells with wild-type IDH2, or in patients plasma or urine. Discussion (or even the Result section) should include comparison of the absolute values and not only ratios towards glutamate (even if this has been a strict condition of the reproducibility study). Authors may comment on conditions under which the activity of wt IDH2 of mutant IDH2 was surveyed. Since 2010 research on the topic has identified numerous factors affecting IDH2 (and IDH1) reactions, namely affecting 2-oxoglutarate as the primary reactant. Distinct contribution of e.g. one-carbon metabolism, state of aerobic glycolysis vs. oxphos and glutaminolysis, etc., are different even in the same immune cells under different conditions. Moreover, enantiomers of 2-hydroxyglutarate have been found to be produced by different enzymes and having distinct metabolism (e.g. dehydrogenases). In the light of these new views, the current reproducibility study represents just a solid beginning in our understanding.

3) Authors should correct the confused labels in Figure 1:

Figure 1 Title legend NADPH −> NADP+ (which actually stands for consumption whereas figure panel A shows the production) is confused with the Figure 1 Title legend. The legends need to be mutually exchanged, so to be for Figure 1 NADP+−> NADPH and for Figure 1 NADPH −> NADP+

---

## [Author Response]

*Essential revisions*

*1) It should be pointed out that specifically precleaning of samples on a column adds to a straightforward analysis. However, it should be verified, whether such precleaning does not withdraw from the sample any desired species to be tested.*

Thank you for raising this point. Correct, sample extraction and clean-up steps may lead to unwanted losses of metabolite levels in samples. For this replication study, we used the exact same methods as the original authors as specified in the Registered Report. The extraction and resin methods are found in other publications by the Thompson group^1,2^. An external validation of the extraction and resin purification methods was not performed as it was outside of the scope of the Replication Study presented here. We have also revised the manuscript to highlight this point.

*2) To bring an additional value besides the simple fact of reproducibility, the authors should discuss why certain other studies have found non zero levels of 2-hydroxyglutarate in cells with wild-type IDH2, or in patients plasma or urine. Discussion (or even the Result section) should include comparison of the absolute values and not only ratios towards glutamate (even if this has been a strict condition of the reproducibility study). Authors may comment on conditions under which the activity of wt IDH2 of mutant IDH2 was surveyed. Since 2010 research on the topic has identified numerous factors affecting IDH2 (and IDH1) reactions, namely affecting 2-oxoglutarate as the primary reactant. Distinct contribution of e.g. one-carbon metabolism, state of aerobic glycolysis vs. oxphos and glutaminolysis, etc., are different even in the same immune cells under different conditions. Moreover, enantiomers of 2-hydroxyglutarate have been found to be produced by different enzymes and having distinct metabolism (e.g. dehydrogenases). In the light of these new views, the current reproducibility study represents just a solid beginning in our understanding.*

We have revised the results/Discussion section to discuss the low levels of 2HG detected in this replication as well as other studies investigating 2HG as a biomarker in AML and other cancers. We did not perform this replication in a manner to provide accurate quantification of 2HG concentrations in the samples because this would have required standard curves, isotope labeled internal standards and recovery experiments. We therefore present semi-quantified data similar the original paper and as outlined in the replication report. However, we did include comments that raw 2HG peak intensities detected in this study followed the trend of 2HG/glutamate ratios.

We also expanded the final section to include that additional biological factors affecting IDH reactions have been published since the original study, specifically differential metabolism of 2HG enantiomers. Specifically, it has been reported that normal tissues can produce *S-*2HG under times of hypoxic stress, which is distinct from *R*-2HG, produced from mutations in IDH1/2. For studies that do not chromatographically separate enantiomers (*R-* vs. *S-* 2HG) both species will give the same *m/z* and spectra, and therefore appear as the same species. It is possible the low levels of *S-*2HG are contributing to the non-zero values. Additionally, depending on type of mass spectral analysis and standard of data processing for each group, the lowest values reported can vary.

*3) Authors should correct the confused labels in Figure 1:*

*Figure 1 Title legend NADPH −> NADP+ (which actually stands for consumption whereas figure panel A shows the production) is confused with the Figure 1 Title legend. The legends need to be mutually exchanged, so to be for Figure 1 NADP+−> NADPH and for Figure 1 NADPH −> NADP+*

Thank you for bringing this error to our attention. We have uploaded the correct version of this figure during resubmission.

1) Wise, D. R. et al. Hypoxia promotes isocitrate dehydrogenase-dependent carboxylation of α-ketoglutarate to citrate to support cell growth and viability. P Natl Acad Sci USA 108, 19611-19616, doi:10.1073/pnas.1117773108 (2011).

2) Lu, C. et al. IDH mutation impairs histone demethylation and results in a block to cell differentiation. Nature 483, 474-U130, doi:10.1038/nature10860 (2012).